# Milk Consumption and Respiratory Function in Asthma Patients: NHANES Analysis 2007–2012

**DOI:** 10.3390/nu13041182

**Published:** 2021-04-02

**Authors:** Stefanie N. Sveiven, Rachel Bookman, Jihyun Ma, Elizabeth Lyden, Corrine Hanson, Tara M. Nordgren

**Affiliations:** 1Division of Biomedical Sciences, School of Medicine, University of California-Riverside, Riverside, CA 92521, USA; ssvei001@ucr.edu (S.N.S.); rachelbookman21@gmail.com (R.B.); 2Biostatistics Department, College of Public Health, University of Nebraska Medical Center, Omaha, NE 68198, USA; jihyun.ma@unmc.edu (J.M.); elyden@unmc.edu (E.L.); 3Medical Nutrition Education Division, College of Allied Health Professions, University of Nebraska Medical Center, Omaha, NE 68198, USA; ckhanson@unmc.edu

**Keywords:** asthma, milk, diet, FEV_1_, FVC, lung, pulmonary, health, asthmatic

## Abstract

Per the Centers for Disease Control and Prevention, asthma prevalence has steadily risen since the 1980s. Using data from the National Health and Nutrition Examination Survey (NHANES), we investigated associations between milk consumption and pulmonary function (PF). Multivariable analyses were performed, adjusted for *a priori* potential confounders for lung function, within the eligible total adult population (*n* = 11,131) and those self-reporting asthma (*n* = 1,542), included the following variables: milk-consumption, asthma diagnosis, forced vital capacity (FVC), FVC%-predicted (%), forced expiratory volume in one-second (FEV_1_), FEV_1_% and FEV_1_/FVC. Within the total population, FEV_1_% and FVC% were significantly associated with regular (5+ days weekly) consumption of exclusively 1% milk in the prior 30-days (β:1.81; 95% CI: [0.297, 3.325]; *p* = 0.020 and β:1.27; [0.16, 3.22]; *p* = 0.046). Among participants with asthma, varied-regular milk consumption in a lifetime was significantly associated with FVC (β:127.3; 95% CI: [13.1, 241.4]; *p* = 0.002) and FVC% (β:2.62; 95% CI: [0.44, 4.80]; *p* = 0.006). No association between milk consumption and FEV_1_/FVC was found, while milk-type had variable influence and significance. Taken together, we found certain milk consumption tendencies were associated with pulmonary function values among normal and asthmatic populations. These findings propound future investigations into the potential role of dairy consumption in altering lung function and asthma outcomes, with potential impact on the protection and maintenance of pulmonary health.

## 1. Introduction

The global burden of pulmonary disease and chronic respiratory diseases (CRD) in particular, are indicated not only by the financial costs of billions of dollars but also in the cost of life by premature mortality of those diagnosed [1,2]. Additionally, as the devastating impacts of COVID-19 have taken hold this past year, we will likely see increased burden of chronic respiratory disease secondary to irreversible pulmonary fibrosis, for years to come [3]. One such CRD, asthma, is often a challenge to control as environmental triggers are typically unavoidable. Furthermore, airway remodeling which may result from chronic, uncontrolled asthma, increases the risk of emphysema and chronic bronchitis. These pulmonary diseases additionally result in increased risk in developing either chronic obstructive pulmonary disease (COPD), another burdensome CRD that is among the top 5 leading causes of death globally, or asthma-COPD overlap syndrome (ACOS) [4,5]. It is anticipated that the US will spend 300 billion dollars on asthma over the next 20 years, with prevalence likely to keep rising as it has over the past decade [6]. While therapeutic agents for asthma are effective in managing symptoms for most individuals, and are dependent on access to long-term treatment both physically and financially [7,8,9]. As yet, asthma remains incurable, and individuals with moderate to severe asthma are reliant upon pharmaceutical intervention to preserve a semblance of normalcy until supportive means that are more accessible and/or a cure are identified [10,11].

Diet is an influential and modifiable factor in disease treatment and prevention which allows for autonomous modulation of disease progression and outcomes, to the extent that environmental factors influence disease. Data suggests that asthma can be exacerbated or remediated by dietary factors. A Western diet, as well as obesity, can contribute to a pro-inflammatory physiological state via excess high saturated-fat intake, and thus serve to exacerbate immune-related diseases like asthma [12,13,14]. Meanwhile, a diet rich in healthy fats, such as the Mediterranean diet, may provide important anti-inflammatory mediators and exert protective effects against inflammatory disease and asthma [15,16,17]. Dairy products contribute a major share of the food supply, but surprisingly little research has been conducted to evaluate the direct effects of milk consumption and fat content on health outcomes like inflammation. Dairy consumption may be protective against asthma also has been associated with improvement in lung function and inflammatory biomarkers in the ECLIPSE cohort of individuals with COPD [18,19]. Low fat, but not high fat, dairy intake has also been associated with a decrease in emphysema, improved lung density measured by computed tomography (CT) images, and a potential inverse relationship between high-fat dairy and FEV_1_/FVC [20]. 

The National Center for Health Statistics (NCHS), a part of the Centers for Disease Control and Prevention (CDC), has conducted a nationwide snapshot of health and disease across the American population, known as the National Health and Nutrition Examination Surveys (NHANES), since the 1950s [21]. These surveys comprise a publicly available data set, accessible online, which aims to provide researchers with epidemiological data for analyses with a variety of data items such as interviews, laboratory tests, physical examinations and other reports. Among data in NHANES, nutritional assessments in which individuals complete validated assessments about their dietary intake, can be sorted in the context of both self-reported and diagnosed disease. Overall, the aim of this dataset is to associate the impacts of long-term exposures and nutritional patterns on health and disease in the United States. 

Using the NHANES 2007–2012 waves, we analyzed milk consumption in participants including those with self-reported asthma to assess for associations between milk consumption and pulmonary function measures among these individuals. 

## 2. Materials and Methods

### 2.1. Study Design

Information about the NHANES cross-sectional study design, as well as the methods of participant selection are publicly available through the CDC.gov website. These studies involved NHANES survey years with both lung function data and asthma questionnaires available. The primary outcomes of this study were self-reported asthma prevalence and lung function measurements. Self-reported asthma was determined using the NHANES questionnaire variables MCQ010 “Ever been told you have asthma: yes/no” and MCQ035 “Still have asthma: yes/no.” Throughout the manuscript, those reporting “yes” on MCQ010 are considered to have a history of asthma, while those reporting “yes” to MCQ035 are considered to have current asthma. In addition, NHANES reports SPXNFVC and SPXNFEV1 provided the data for analysis of lung function outcomes. The NCHS acquired these data during home health interviews and mobile examination centers (MEC) to obtain laboratory data. NHANES waves 2007–2012 were used since pre-bronchodilator spirometry was performed these years. The outcomes used to assess lung function included: forced expiratory volume in one second (FEV_1_), forced vital capacity (FVC), FEV_1_/FVC ratio, FEV_1_ percent predicted, and FVC percent predicted. Protocols for these measurements are also summarized within the NHANES resources. Lung function was expressed as a percent predicted based on age, gender, height and race (white, black or Mexican, Hispanic, or other) from the third NHANES reference values [22]. 

### 2.2. Participants

Participants aged 19 to 79 with pre-bronchodilator spirometry data (FEV_1_ and FVC) quality grades A and B were used in the analysis. The NHANES Cohort for waves 2007–2012 meeting these criteria totaled 30,442. After applying exclusion criteria—pregnant women, and those with energy intake greater than or less than the plausible intake (<600 or >6000 kcal/day for adult females, 800–8000 kcals/day for adult males)—the remaining eligible participants totaled 11,180. Individuals lacking self-reported asthma status were removed from the dataset, making the final participants for the general population of the study equal 11,131. Of this population, those indicating a history of asthma and/or current asthma totaled 1, 542. These details and a flowchart following STROBE (Strengthening the Reporting of Observational Studies in Epidemiology) principles are provided in the graphical abstract. NHANES cycles between 2007–2012 were combined with multi-wave weight adjustments. Details for the NHANES study are described within the website reference [21].

### 2.3. Variables 

Milk consumption in NHANES includes the following self-reported variables: any type of milk 5x weekly; milk products consumed <30 days; whole (full-fat) milk consumed <30 days; 2% (reduced fat) milk consumed <30 days; 1% (low fat) milk consumed <30 days; skim milk (fat free) consumed <30 days; whole milk only consumed <30 days; 2% milk only consumed <30 days, 1% milk only consumed <30 days, skim milk only consumed <30 days, 2+ types of milk consumed <30 days. The terms of our analyses relating to milk consumption in the NHANES population are defined in Table 1. Outcome variables assessed included pulmonary function values measured by spirometry (FEV_1_, FVC, FEV_1_ percent predicted [FEV_1_%], FVC percent predicted [FVC%], FEV_1_/FVC), milk consumption, and self-reported asthma (history of and/or current). Pulmonary function tests are diagnostic for lung diseases including COPD, emphysema, and asthma. The most clinically useful values are FEV_1_/FVC which may be indicative of obstructive lung disease when <0.70, a category to which some asthma patients may ascribe. However, this ratio is neither sensitive enough nor accurate enough for diagnosis [23]. FEV_1_% and FVC% values provide clinical information within the context of factors known to impact lung function (e.g., height, age, and gender). 

Potential confounding variables were chosen a priori based on previous associations found in literature and included gender, race, education, smoking status, body mass index (BMI) group, previous work exposure to mineral dusts, previous work exposure to natural dusts, and poverty index [24,25,26,27,28,29]. Cut-offs for poverty were defined by the NHANES survey variable “INDFMPIR,” calculated as a ratio of individual/family income to poverty guidelines determined by the Department of Health and Human Services with a range from 0–5. Values from 0–1.35 were considered “poor”, 1.36–1.85 were considered “nearly poor”, and values of 1.86+ were considered “not poor” for classification. Mineral dust and natural dust exposure data were determined by the NHANES questionnaire as a response to variable OCQ510, “Ever had work exposure to mineral dusts?”, and variable OCQ530, “Ever had work exposure to organic dusts?”. Additionally, the use of the term ‘gender’ here as opposed to ‘sex’ is in accordance with the NHANES use of ‘gender’ to describe male or female respondents. The World Health Organization has standardized the following BMI categories which were used in this study: underweight: <18.5, normal range: 18.5–24.9, overweight: 25–29.9, and obese: ≥ 30. The Centers for Disease Control and NCHS define smoking status as follows: Never (has never smoked, or who has smoked less than 100 cigarettes in their lifetime), current smoker (has smoked 100 cigarettes in their lifetime and who currently smokes cigarettes), or former (has smoked at least 100 cigarettes in their lifetime but who had quit smoking at the time of interview). 

### 2.4. Statistical Methods

Descriptive statistics included counts and percentages for categorical data and means for continuous data to describe the study population from the NHANES cohort for the years 2007–2012 (Table 2). Chi-square test for categorical variables and ANOVA test for continuous variables were applied to examine the characteristics of the demographic, socio-economic, and risk factors related to pulmonary function with participants’ current and past asthma. To evaluate associations between milk consumption and pulmonary function and current asthma status, multivariable linear and binomial logistic regression analyses were conducted with other demographic, socio-economic, and risk factor variables. Diagnostics and fit statistics for the regression analyses were reviewed to assess the validity of our multivariable models. The overall diagnostics were satisfied with regression assumptions and fit statistics significant or acceptable, as appropriate (Wald chi-square <0.0001 for all logistic regressions; diagnostic plots showed no evidence against normality, linearity and equal variances for linear regressions, respectively). A two-sided significance cutoff was set to *p* < 0.05. SAS version 9.4 was used, specifically SAS procedures, “PROC SURVEYFREQ”, “PROC SURVEYMEANS”, “PROC SURVEYLOGISTIC” and PROC SURVEYREG” were used in computing descriptive and regression analyses as these protocols account for both the weighted data as well as the complexity of sample design.

## 3. Results

### 3.1. Descriptive Data of Eligible Participants among the Nhanes Cohort

For these analyses, the number of total eligible participants was 11,180. Of those participants, 49 were missing self-reported asthma status withdrawn from the eligible participants. Of the remaining 11,131 participants, 49.0% were male and 51.0% female. These eligible participants had a mean age of 44.4 years with 54.8% of participants reporting as non-smokers, 23.8% as former smokers, and 21.4% as current smokers. Participant demographics and milk consumption variables were assessed among the total eligible participants and provided in Table 2. 

### 3.2. Milk Consumption Tendencies and Lung Function Measurements in All Eligible Participants

Initial univariable analysis in this total population (*n* = 11,131) identified several significant associations with asthma diagnosis, including age, gender, race, BMI, poverty status, regular milk-drinker (5+ days per week) status across a lifetime, and measurements of lung function (Baseline FEV_1_ and FVC, FEV_1_% predicted and FVC% predicted, as well as FEV_1_/FVC ratio), as identified in Table 2. In accordance with our hypothesis that milk consumption is associated with better lung function, we explored whether milk consumption was associated with differences in lung function parameters in the total eligible population. As shown in Table 3, multivariable regression models demonstrated that lifetime regular milk consumption was significantly associated with FEV_1_ (overall *p* = 0.004), where being a lifetime regular milk drinker (β:54.5; *p* = 0.001) or reporting to sometimes be a regular milk drinker throughout life versus never (β:58.4; *p* = 0.006) was associated with significantly higher FEV_1_ measurements compared to individuals identifying as never being regular milk drinkers. In addition, reporting milk consumption often in the past 30 days was also associated with increased FEV_1_ (β:39.6; *p* = 0.036). An increase in FEV_1_% was determined in individuals identifying as only regularly drinking 1% milk versus no milk (β:1.81; *p* = 0.020) in the prior 30 days.

Similar to findings with FEV_1_, lifetime regular milk consumption was also significantly associated with measured FVC (*p* = 0.018), where identifying as a lifetime regular milk drinker (β:57.7; *p* = 0.001) or sometimes being a regular milk drinker across life (β:58.8; *p* = 0.011) versus never was significantly associated with having higher FVC measurements. While FEV_1_/FVC was significantly associated with asthma diagnosis in the univariable analysis shown in Table 2, there were no significant associations identified with milk consumption tendencies (all *p* values > 0.05). 

### 3.3. Milk Consumption Tendencies and Current Asthma Report in All Eligible Participants

When considering regular, 5+ days per week, milk consumption, multivariable analysis between regular milk consumption across a lifetime and likelihood of reporting current asthma (responding ‘yes’ to “still have asthma”), no significant association was identified (Figure 1A). However, when dichotomized (Yes/No) for being a lifetime regular milk drinker, there was a significant association (OR: 0.81; *p* = 0.026) between identifying as a lifetime regular milk consumer and decreased likelihood of having current asthma. Regular milk consumption in the past 30 days (regardless of type) did not have any significant association with current asthma (Figure 1B). Although, reporting regular milk consumption of exclusively ‘other’ milk (e.g., soy) was significantly associated with reduced current asthma (OR: 0.51 [0.28, 0.93]; *p* = 0.028; Figure 1C).

### 3.4. Milk Consumption Tendencies and Current Asthma Report in Participants Reporting Asthma (History or Current)

Limiting our analysis to individuals specifically reporting a history of asthma or current asthma, Table 4 outlines the characteristics of these asthmatic participants. The total number of participants self-reporting either a history of previous asthma or current asthma was 1542. In this subset of individuals, multivariable analyses identified significant associations between individuals answering ‘yes’ to “still have asthma” versus answering ‘no’, including age, gender, poverty status, and/or lifetime regular milk consumption status. 

Amongst patients self-reporting a history of asthma, we identified a significant association between lifetime regular milk consumption and current asthma status (overall *p* = 0.006). As shown in Figure 2A, those who reported being a lifetime regular milk consumer had a decreased likelihood of reporting current asthma compared to those who reported never being a regular milk drinker (OR: 0.75; 95% Confidence Interval [0.56, 1.01]), while individuals reporting variable regular milk consumption did not exhibit the same effect (OR: 1.13; 95% CI: [0.76, 1.69]). When this variable was dichotomized, we similarly identified a significant association (*p* = 0.001), where individuals that were identified as regular milk drinkers were significantly less likely to report current asthma compared to individuals not identified as being regular milk drinkers (OR: 0.70; 95% CI: [0.56, 0.86]). 

When considering milk consumption (of any milk type) over the 30 days prior to survey participation, we identified no significant association between milk consumption and reporting current asthma (overall *p* = 0.100). Shown in Figure 2B, as compared to individuals never drinking milk in the past 30 days, consumers reporting once daily or more milk consumption had reduced odds of reporting current asthma (OR: 0.62 [0.40, 0.95]; *p* = 0.028). Those reporting sometimes or rarely did not exhibit a similar association. When this variable was dichotomized to those reporting daily milk consumption in the past 30 days versus those not reporting daily milk consumption, there was no significance (OR: 0.76; 96% CI: [0.56, 1.02]; *p* = 0.063). Furthermore, as shown in Figure 2C, when assessing regular, exclusive consumption of a specific milk type in the prior 30 days, individuals reporting exclusive consumption of whole milk only versus no milk had a significantly decreased likelihood of reporting current asthma (OR:0.60; 95% CI: [0.38, 0.96]; *p* = 0.032). 

### 3.5. Milk Consumption Tendencies and Lung Function Parameters in Participants Reporting Asthma (History or Current)

As in our assessment of the general population, we also evaluated for relationships between lung function and milk consumption in the cohort reporting current/history of asthma (Table 5). Here, we found no significant association between identifying as a regular milk consumer across lifetime and any differences in FEV_1_ or FEV_1_% measures.

When looking at FVC, lifetime regular milk consumption tendencies and FVC measurements did not reach significance (overall *p* = 0.058). Although, individuals who reported sometimes being a regular milk drinker across their life had significantly higher FVC measurements compared to participants who did not identify as ever being a regular milk drinker (β:127.28; *p* = 0.030). This finding is paralleled by the FVC% findings of a significant association between lifetime regular milk consumption tendencies and FVC% (overall *p* = 0.045) with those reporting as variably being regular milk drinkers having significantly higher FVC% (β:2.62; *p* = 0.020) vs. those identifying as never being a regular milk consumer. Meanwhile, exclusive consumption of specific types of milk in the prior 30 days were also associated with significant differences in FVC outcomes (overall *p* = 0.0084) and FVC% outcomes (overall *p* = 0.009). Here, consumption of only skim milk the 30 days prior to completion of the questionnaire (β:2.76; *p* = 0.045) was associated with higher FVC% measurements compared to those consuming no milk in the prior 30 days.

## 4. Discussion

Asthma has been on the rise across the world [30], and dietary modification is a viable option for reducing incidence of respiratory diseases like asthma. Lately, there is growing awareness about the role of dairy products in health, with studies identifying that intake of dairy products does not increase the risk for disease and may be protective [31,32,33,34]. Our findings suggest that milk consumption may be an autonomous factor that individuals can modify to benefit lung health outcomes with consumption reducing instances of current asthma reports in those with and without a history of asthma. In these investigations, we have utilized the NHANES 2007–2012 dataset to assess relationships between milk consumption tendencies and asthma-related outcomes, including reporting current asthma as well as lung function measurements. Since these data are generated from a cross-sectional study, the FEV_1_% and FVC% are more informative to pulmonary performance since they are inherently compared to expected values. Among the total population, we found FEV_1_% and FVC% to be significantly associated with regular consumption of exclusively low-fat 1% in the prior 30-days (β:1.81; *p* = 0.020 and β:1.27; *p* = 0.046). Among the total population, we also identified significant associations with FEV_1_ (*p* = 0.0040), FVC (*p* = 0.018), and regular milk consumption. There was a significant association between FEV_1_ and FVC in participants describing regular milk consumption as lifetime regular consumption (FEV_1_ β: 54.5; *p* = 0.0013 and FVC β:57.7; *p* = 0.010) or varied regular consumption ({FEV_1_, β:58.4; *p* = 0.006 and FVC, β:58.8; *p* = 0.011). 

Among participants with asthma, varied-regular milk consumption in a lifetime was significantly associated with FVC (*p* = 0.002) and FVC% (*p* = 0.006). These paralleled findings may be indicative of the importance of timing in milk consumption as we explore below, in which regular milk consumption earlier in life is most strongly associated with changes in lung function [35,36,37]. Among these participants, individuals who were regular milk drinkers during childhood, as many parents transition children to cow’s milk in the earlier years, may have ceased frequent consumption in adulthood. This is an example warranting a “varied/sometimes” response to lifetime regular milk consumption, where the beneficial effects of cow’s milk during earlier years could provide protection to pulmonary function into adulthood. In our investigations, FVC and FVC% of participants with asthma were additionally associated with regular single-type milk consumption in the prior 30-days (*p* = 0.008 and *p* = 0.009, respectively) with regular consumption of exclusively low-fat 1% in the prior 30-days associating significantly with FVC% (β:2.76; *p* = 0.045). These data have revealed that individuals identifying as lifetime regular milk consumers are less likely to answer *yes* when asked if they have had or still have asthma and also had higher FEV_1_ and FVC measurements compared to individuals not identifying as a lifetime regular milk drinker. Meanwhile, individuals reporting a history of asthma did not have any associations with lifetime milk consumption and FEV_1_. We identified a potential protective association for exclusive consumption of low fat (2%, 1%, or skim) milks including reduced likelihood of current asthma or higher lung function measurements in both the general and asthmatic populations. Together, these data from NHANES support additional findings in the literature on the effects of dairy consumption and lung health, particularly in the case of low-fat milks. We cannot make causative claims based on these results, but in summation with similar findings in the literature, we can appreciate the potential impact of dairy consumption on pulmonary health.

Varied results in our findings may be the consequence of the inability to assess milk quality and content in these individuals. Many studies have clarified the impact animal well-being has on milk components. In particular, heat-stress resulting from challenging temperature conditions exhibits detrimental effects to the animals’ physiological state, altering the quantity and quality of milk products. It has been demonstrated that heat stress results in a decrease in yield, as well as fat and protein milk-content, and likely aberrant nutritional value [38,39]. Furthermore, significant changes to the triacylglyceride composition (decreased SCFA, MCFA, lipid polar classes and increased LCFA) of milk due to acute heat stress presumably alters the biological properties of milk [40]. Alternatively, transportation stress demonstrated an increase in fat content, when comparing milk from transported to non-transported cows, and there were significant decreases in pH, yield, as well as lactose and solid non-fat content associated with transportation [41]. Transportation stress was also significantly associated with increased leukocyte, neutrophil, eosinophil, and monocyte count, although cytokines were not measured. Additionally, breed and genetics of cows play an important role in the quality of milk produced, particularly as this field is heavily reliant upon SNP analysis and sequencing screening [42,43,44]. The properties of milk are likely to differ between dairy farms, as well as the brands that are available at participants’ grocery stores. Thus, we anticipate these factors to be influential in our reported results and we suggest further studies investigating how they alter the biological impacts of dairy consumption on human lung physiology and asthma.

Finally, protocols for milk processing (pasteurization, homogenization, milk-fat removal) modify the properties of milk. Most commonly, milk consumed in the United States is treated with ultra-high temperatures for pasteurization and homogenization, and therefore the impact of milk consumption is dependent on milk-processing. The literature on the benefits of raw milk in human physiology, immunology, and protection against asthma, is abundant. Since raw milk is not heat-processed, the protein structures remain intact as well as the microbial diversity. The GABRIEL Advanced study demonstrated, using a questionnaire and blood samples when available, that among 79,888 school-aged children living in rural Austria, Germany and Switzerland, both raw and pasteurized milk consumption may half the risk of developing asthma [45]. A literature review by Barbara Sozánska M.D. summarized that consumption of raw cow’s milk is protective against allergies and asthma among children and adults [31]. These studies demonstrate the influence that dairy consumption may have specifically on lung health, via a variety of mechanisms, including dietary absorption of lipids, macronutrients, or milk-derived exosomes containing small regulatory RNAs and immunoglobulins. The fat-soluble vitamins in milk, D and E, have promising if not well-established associations with improved inflammatory processes, and specifically in the airways [46]. Vitamin D deficiency, for example, results in abhorrent lung structure and function in a mouse model [47]. Vitamin D deficient mice have significantly less thoracic gas volume, greater airway resistance, and lower alveolar air volume and less alveoli. In human, insufficient serum vitamin D is associated with asthma severity markers such as elevated IgE, eosinophilia, and increased hospitalizations [48]. A meta-analysis identified a direct relationship between vitamin D levels and FEV_1_, FEV_1_%, and FEV_1_/FVC [49]. Additionally, vitamin D plays an important role in inhibiting human airway smooth muscle cells, which is an important measure of asthma severity [50].

Dairy fat is comprised mainly of long, medium, and short-chain saturated fatty acids (LCFA, MCFA, SCFA). Certain long-chain fatty acids have been associated with increased serum lipid levels, and corresponding risk for cardiovascular disease, however milk has a much higher proportion of short- and medium chain saturated fatty acids. These varying chain lengths impact blood lipid levels and inflammation differently and may be more important than previously recognized. Evidence suggests these short-chain fatty acids have potential anti-inflammatory effects due to their influence on the regulation of various molecular signaling pathways [51], and around 11% of the fat content of milk is SCFA [52]. Studies suggest that intake of dairy products, high fat or raw, lowers inflammatory markers in adults and increases the intake of the anti-inflammatory ω-3 polyunsaturated fatty acid in children, respectively [53,54]. These findings, along with findings defining alterations to milk properties, demonstrate the impact milk quality may have on lung health and support observations found in the current study.

When considering the potential mechanisms by which milk elicits its effects on human lung health, it is important to consider the many homologous components of bovine milk such as immunoglobulins, lactoferrin, lactadherin, and various cytokines that may enhance the immune activity upon consumption and absorption [54]. Low-fat dairy intake also has implications in positive clinical outcomes of moderately improved lung density, measured by CT scan, the mechanisms of which were not fully elucidated [20]. Furthermore, current consumption of farm milk leads to a significant increase in regulatory T cell counts and potential increase in activated T cells [55]. Furthermore, we have previously shown that mice fed a diet containing bovine milk exhibited altered immune responses to an environmental dust challenge, compared to mice fed the same diet where the milk was first sonicated to disrupt milk exosomes. Here, mice fed the unsonicated milk diet exhibited a skewing towards an M1-type inflammatory response, while mice fed the sonicated milk diet exhibited a polarization towards an M2-like response that is classically seen in asthma and allergic disease settings [56]. Diet-derived small RNAs are also implicated in gene expression regulation at the level of the consumer. The analogous nature of many miRNAs, from bovine to human, implicates the role of milk-derived non-coding RNAs in targeting genes related to inflammation including in disease settings such as asthma [54,56,57]. Therefore, lung function and immunity may be modulated by dairy consumption.

In a cohort of men and women, aged 45–84 years, the MESA-Lung Study questionnaire demonstrated the significance of higher low-fat dairy consumption on improved CT-measured lung density [20]. The implications of this study are that dairy consumption increases tissue to air ratio as a measure of improvement of the expanded airways and alveolar tissue destruction as found in emphysema. Additionally, pulmonary function testing is a clinical standard for diagnosing and managing obstructive lung disease like asthma [58]. While no CT data were available for this NHANES cohort, spirometry can be used to ascertain improvements in lung health. In this cohort, milk consumption overall was associated with higher pulmonary function, measured by spirometry, in asthmatics. Among those reporting asthma, those that reported sometimes or regularly consumed dairy products across life and/or within the prior 30 days had overall better FEV_1_/FVC measurements, with the exception of individuals reporting regular whole milk consumption in the prior 30 days (as mentioned above).

It is worth mentioning that people with asthma can be avoidant of dairy products because of what is known as the “milk-mucus theory.” This theory suggests that consumption of dairy products exacerbates asthma symptoms, but the summation of data from a wealth of studies does not directly link milk consumption to asthma, and overall there is no evidence of cause and effect [59]. There are studies in which people with asthma reported improved respiratory outcomes when eliminating dairy consumption, however a number of studies demonstrated that placebo control also reported improved lung function [60,61]. While the evidence does not support the milk-mucus theory, it is important to be aware of this when educating patients/individuals with asthma about any recommendations of incorporating dairy products. 

The results reported through our investigations and derived from a cross-sectional study provide an interesting, although limited, perspective on the impact of dairy consumption and pulmonary health outcomes. For example, there are likely unmodelled confounding variables and interactions that we have not assessed. Additionally, NHANES relies on participants self-reporting asthma, thus this work lacks quantitative data of asthma diagnoses, symptomology, and potential ongoing therapeutics usage. Due to the exploratory nature of these studies in assessing for significant associations between milk consumption and pulmonary function/disease outcomes, we did not apply any multiplicity corrections. In addition, though frequency of milk consumption was described, there were no data available regarding the quantity of milk consumption, thus quantity is not effectively modelled, which is likely to impact these outcomes. Beyond milk consumption, it would be beneficial to study if dairy consumption in forms of cheese and yogurt are associated with lung function and/or asthma in the NHANES dataset, or if participants differentiated between these products. These limitations could be addressed through further studies focused on elucidating how frequency, quantity, and type of dairy consumption alter pulmonary outcomes across a period of time, to better understand the role of milk in improving lung health outcomes. In particular, identifying how soon after milk consumption habits might elicit alterations in pulmonary outcomes would be important if dairy products are to be recommended for lung health. To address the shortcomings of this cross-sectional study design, further investigation using reversibility testing, highly sensitive for asthma, could also be used. All together, these data in combination with previously published data, highlight the importance of further studies on dairy and its potential to modify lung disease outcomes, prevent disease, and use in disease treatment where current access to therapeutics falls short. 

## Figures and Tables

**Figure 1 nutrients-13-01182-f001:**
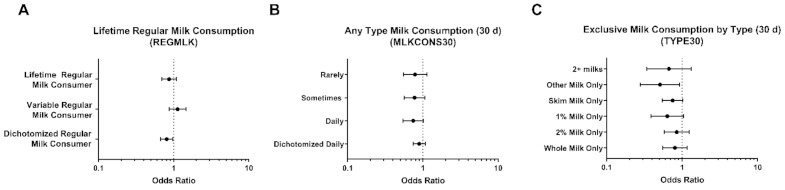
Associations between milk consumption tendencies and reporting current asthma in the NHANES total population. Odds ratios for reporting current asthma compared to the total NHANES population based on identifying as (**A**) a lifetime regular milk drinker, or sometimes being a regular milk drinker across life as compared to identifying as never being a regular milk drinker; (**B**) based on any type milk consumption tendencies in the previous 30 days; (**C**) based on exclusive regular milk consumption by type.

**Figure 2 nutrients-13-01182-f002:**
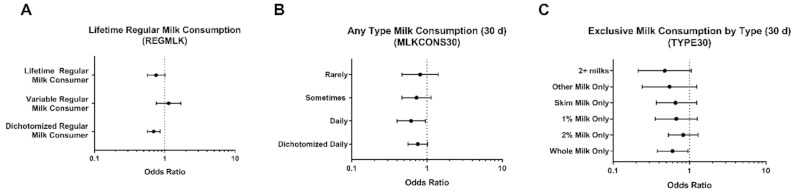
Associations between milk consumption tendencies and reporting current asthma in participants reporting asthma (history or current). Odds ratios for reporting current asthma amongst individuals reporting asthma based on identifying as (**A**) a lifetime regular milk drinker, or sometimes being a regular milk drinker across life as compared to identifying as never being a regular milk drinker; (**B**) based on any type of milk consumption tendencies in the previous 30 days; (**C**) based on regular (5+ days per week) exclusive type of milk consumption.

**Table 1 nutrients-13-01182-t001:** Definition of terms used in analyses.

Consumption Frequency Variables	Definition of Variable	Response Categories
Lifetime regular milk drinker (REGMLK)	Milk consumption, regardless of type, 5+ days per week for a lifetime	Lifetime: most of their life Varied: sometimes a regular milk drinker Never: never a regular milk drinker
Dichotomized lifetime regular milk drinker (REGMLKR)	Milk consumption, regardless of type, 5+ days per week for a lifetime dichotomized as Lifetime/Varied or Never	Yes (Lifetime) No (Varied/Never)
Any type milk consumption within prior 30 days (MLKCONS30)	Frequency of milk consumption, regardless of type, in the 30 days prior to survey date	Daily: 1+ times daily Sometimes: 1+ times a week but not daily Rarely: less than 1 time per week Never: no milk consumption
Dichotomized any type milk consumption within prior 30 days (MLKCONS30R)	Frequency of milk consumption, regardless of type, in the 30 days prior to survey date, dichotomized as 5+ times a week or Sometimes/Rarely/Never	Yes (5+ times a week or more) No (Sometimes/Rarely/Never)
**Type of Milk Consumption Variables**	**Definition of Variable**	**Response Categories**
Exclusive milk consumed by type 5+ days per week in prior 30 days (TYPE 30)	Whole (full fat) only, 2% (reduced fat) only, 1% (low fat) only, skim (fat free) only, other type of milk (non-dairy, such as soy) only, 5+ days per week in last 30 days	Yes No

Variables used throughout the study are separated into two categories (bolded): consumption frequency and consumption type. Each variable, left justified in the first column, is followed by a definition and then reported with associated response categories.

**Table 2 nutrients-13-01182-t002:** Characteristics of NHANES participants stratified by self-reported asthma status.

Characteristic	ALL (*n* = 11,131)	No History of Asthma (*n* = 9589)	Yes, Have a History of Asthma (*n* = 717)	Yes, Still Have Asthma (*n* = 825)	*p*-Value
	Mean or %	Mean or %	Mean or %	Mean or %	
Age (year)	44.4	44.8	40.2	44.0	**<0.0001**
Gender					**<0.0001**
Female	51.0%	49.6%	53.4%	64.2%	
Male	49.0%	50.4%	46.6%	35.8%	
Race					**<0.0001**
Hispanic	13.6%	14.2%	11.1%	8.7%	
Non-Hispanic White	69.9%	69.5%	70.6%	74.0%	
Non-Hispanic Black	10.3%	10.0%	11.9%	12.8%	
Other Race	6.2%	6.3%	6.4%	4.5%	
BMI group					**0.0002**
Underweight/Normal: < 25.0	31.5%	32.0%	31.4%	26.4%	
Overweight: 25.0–29.9	33.5%	34.1%	29.0%	30.2%	
Obesity: > 30	35.0%	33.9%	39.6%	43.4%	
Smoking					0.250
Non-Smoker	54.8%	55.2%	54.9%	50.4%	
Former Smoker	23.8%	23.7%	23.7%	24.5%	
Smoker	21.4%	21.1%	21.4%	25.1%	
Poverty					**0.0004**
Poor	21.4%	20.7%	22.8%	28.0%	
Nearly poor	8.8%	8.6%	8.9%	10.6%	
Not poor	69.8%	70.6%	68.3%	61.4%	
Education					0.230
Less than 12th grade	15.8%	16.0%	14.6%	14.4%	
High school/GED	22.0%	22.0%	20.7%	23.5%	
Some college or AA	32.0%	31.4%	34.7%	35.6%	
College or above	30.2%	30.5%	30.1%	26.5%	
Mine dust					0.480
No	68.2%	68.0%	70.3%	69.7%	
Yes	31.7%	32.0%	29.7%	30.3%	
Natural dust					0.500
No	76.9%	79.2%	81.1%	75.7%	
Yes	23.1%	20.8%	18.9%	24.3%	
Regular Milk Drinker					0.049
Never been	21.3%	21.2%	20.9%	23.1%	
Sometimes/varied	35.8%	35.8%	31.6%	38.8%	
Lifetime	42.9%	43.0%	47.5%	38.1%	
Lifetime regular milk drinker (Yes/No)					0.016
Yes	45.2%	46.0%	38.5%	41.4%	
No	54.8%	54.0%	61.5%	58.6%	
Any type milk consumption (30 days)					0.410
Never	15.8%	15.7%	14.3%	18.2%	
Rarely: < once a week	15.4%	15.5%	13.8%	15.9%	
Sometimes: < once a day	29.8%	30.0%	28.7%	28.3%	
Often: once a day or more	39.0%	68.8%	43.2%	37.6%	
Any type milk consumption 5X a week or more (30 days)					0.160
Yes	61.0%	61.2%	56.8%	62.4%	
No	39.0%	38.8%	43.2%	37.6%	
Exclusive type milk consumed (30 days)					0.720
Never	15.8%	15.7%	14.3%	18.2%	
Regular/whole milk only	17.4%	17.3%	19.0%	16.3%	
2% milk only	31.5%	31.4%	30.9%	33.1%	
1% milk only	12.0%	12.1%	10.4%	11.1%	
Skim milk only	16.5%	16.4%	17.5%	15.7%	
Other milk only	3.9%	4.0%	3.6%	2.6%	
2+ types milk combined	3.1%	3.0%	4.2%	3.0%	
Baseline FEV_1_ (mL)	3245.6	3273.9	3317.1	2868.2	**<0.0001**
Baseline FVC (mL)	4155.4	4155.4	4221.7	3841.3	**<0.0001**
FEV_1_% predicted	96.3	97.1	95.3	88.7	**<0.0001**
FVC % predicted	99.1	99.5	98.8	96.0	**<0.0001**
FEV_1_/FVC	0.78	0.78	0.78	0.75	**<0.0001**

*p*-values represent the Wald-type/overall *p*-values associated with the history of asthma (Yes, have a history and Yes, still have asthma) against no history of asthma. Bolded *p*-values represent those reaching statistical significance at *p* ≤ 0.5. Column headings at the top define the content of each column. Response categories for each characteristic are indented below the corresponding characteristic. Definitions: body mass index (BMI); General educational development (GED) signifying high-school level academic skills; Associate in Arts (AA) degree; forced expiratory volume in one-second (FEV_1_); forced expiratory volume in one-second (FVC).

**Table 3 nutrients-13-01182-t003:** Multivariable models of associations between milk consumption tendencies and lung function measurements in the total NHANES population.

Milk Consumption Tendencies (*n* = 11,180)	FEV_1_		FVC		FEV_1_%		FVC%		FEV_1_/FVC	
	β ^1^ [SE] ^2^	*p* ^3^	β ^1^ [SE] ^2^	*p* ^3^	β ^1^ [SE] ^2^	*p* ^3^	β ^1^ [SE] ^2^	*p* ^3^	β ^1^ × 10 ^3^ [SE × 10^3^] ^2^	*p* ^3^
Regular Milk Drinker (REGMLK vs. Never)		**≤0.01**		**0.02**		0.17		0.29		0.69
Lifetime	54.54 [16.0]	***≤0.01***	57.71 [21.4]	***≤0.01***	0.59 [0.44]	*0.19*	0.32 [0.39]	*0.42*	1.78 [2.54]	*0.49*
Variable/Sometimes	58.42 [20.0]	***≤0.01***	58.77 [22.3]	≤0.01	1.05 [0.54]	*0.06*	0.64 [0.38]	*0.10*	3.04 [3.48]	*0.39*
30-Day Milk Consumption Frequency (MLKCONS30 vs. Never)		0.20		0.32		0.35		0.47		0.36
Rarely	26.34 [23.0]	*0.26*	13.88 [25.0]	*0.58*	0.95 [0.63]	0.14	0.40 [0.50]	*0.43*	4.18 [2.87]	*0.63*
Sometimes	30.44 [23.7]	*0.20*	22.11 [27.6]	*0.42*	0.97 [0.61]	*0.11*	0.64 [0.50]	*0.21*	3.60 [3.07]	*0.15*
Often	39.55 [18.3]	***0.04***	40.93 [22.4]	*0.07*	0.92 [0.57]	*0.12*	0.73 [0.48]	*0.14*	1.40 [2.90]	*0.25*
Regular 30-day Exclusive Type Milk Consumption (TYPE30 vs. Never)		0.14		0.30		0.16		0.07		0.11
Whole	−1.10 [21.7]	0.96	−5.93 [24.1]	0.81	0.29 [0.58]	0.62	0.12 [0.47]	0.79	0.44 [3.29]	0.32
2%	36.34 [19.8]	*0.07*	37.02 [24.4]	*0.14*	0.92 [0.56]	*0.11*	0.81 [0.45]	*0.08*	1.72 [3.01]	*0.57*
1%	59.50 [32.7]	*0.08*	52.02 [39.2]	*0.19*	1.81 [0.75]	***0.02***	1.27 [0.62]	***0.05***	5.07 [3.15]	*0.11*
Skim	50.41 [25.5]	*0.05*	45.96 [29.4]	*0.12*	1.28 [0.74]	*0.09*	0.83 [0.60]	*0.18*	3.60 [3.61]	*0.32*
Other	30.50 [35.6]	*0.40*	12.63 [44.8]	*0.78*	0.65 [0.99]	*0.51*	−0.03 [0.94]	*0.97*	6.38 [3.56]	*0.08*
2+ types	32.72 [42.7]	*0.45*	6.15 [48.4]	*0.90*	0.09 [1.01]	*0.93*	−0.59 [0.91]	*0.52*	6.28 [4.93]	*0.20*

^1^ Regression coefficient (β); Each regression model shows milk consumption adjusted for the following confounding factors: age, gender, race, education, poverty level, BMI group, smoking status, mineral dust exposure in the past, and natural dust exposure in the past. ^2^ Standard Error (SE); Due to small values of regression coefficient and standard error, values were displayed after multiplying by 10^3^. ^3^
*p*-values ≤ 0.05 are bolded and have been rounded to the nearest hundredth with ≤ 0.01 for any values less than 0.01; data rounded to the nearest hundredth (greater than or equal to 0.005 rounded up) with ≤0.01 for any values less than 0.01; Non-italicized *p*-values (top *p* value for each variable) represent Wald-type/overall *p*-values. Column headings at the top define the content of each column. Response categories for each characteristic are indented below the corresponding characteristic. Definitions: forced expiratory volume in one-second (FEV_1_); forced expiratory volume in one-second (FVC).

**Table 4 nutrients-13-01182-t004:** Characteristics of milk drinkers and association to symptoms among participants reporting a history of asthma (*n* = 1542).

Characteristic	Current Asthma	Current Asthma
	Yes vs. No	Yes vs. No
	Univariable	Multivariable
	*p* value ^1^	*p* value ^1^
	OR ^2^ [95% CI] ^3^	OR ^2^ [95% CI] ^3^
Age (year)	***p* < 0.001**	***p* = 0.001**
	1.02 [1.01, 1.02]	1.02 [1.01, 1.03]
Gender	***p* = 0.001**	***p* = 0.029**
Female	1.56 [1.21, 2.02]	1.43 [1.04, 1.96]
Male	1	1
Race	*p* = 0.107	*p* = 0.319
Hispanic	0.74 [0.52, 1.07]	0.74 [0.51, 1.08]
Non-Hispanic White	1	1
Non-Hispanic Black	1.03 [0.76, 1.40]	0.96 [0.68, 1.37]
Other race	0.67 [0.42, 1.06]	0.71 [0.40, 1.25]
BMI group	*p* = 0.229	*p* = 0.893
Underweight/Normal: > 25.0	1	1
Overweight: 25.0-29.9	1.25 [0.87, 1.78]	1.10 [0.75, 1.61]
Obesity: > 30	1.30 [0.96, 1.77]	1.08 [0.73, 1.60]
Smoking	*p* = 0.257	*p* = 0.600
Non-smoker	1	1
Former smoker	1.13 [0.83, 1.54]	0.99 [0.69, 1.41]
Smoker	1.28 [0.96, 1.70]	1.17 [0.85, 1.62]
Poverty	*p* = 0.120	***p* = 0.044**
Poor	1.37 [1.00, 1.87]	1.47 [1.04, 2.08]
Nearly poor	1.32 [0.84, 2.10]	1.41 [0.92, 2.14]
Not poor	1	1
Education	*p* = 0.652	*p* = 0.637
Less than 12th grade	1	1
High school/GED	1.15 [0.76, 1.75]	1.34 [0.82, 2.18]
Some college or AA	1.04 [0.71, 1.52]	1.29 [0.83, 2.00]
College or above	0.89 [0.60, 1.33]	1.13 [0.71, 1.81]
Mine dust	*p* = 0.791	*p* = 0.085
No	1	1
Yes	1.03 [0.83, 1.28]	1.27 [0.97, 1.67]
Natural dust	*p* = 0.083	0.051
No	1	1
Yes	0.83 [0.66, 1.03]	0.77 [0.59, 1.00]
Regular milk drinker	***p* = 0.002**	***p* = 0.006**
Lifetime	0.73 [0.57, 0.93]	0.75 [0.56, 1.01]
Sometimes/Varied	1.12 [0.78, 1.60]	1.13 [0.76, 1.69]
Never been	1	1

^1^ Note: Wald-type/overall *p*-values ≤ 0.05 are bolded; ^2^ Odds ratio (OR); ^3^ 95% Confidence Interval (95% CI); percents rounded to the nearest tenth; OR, CI, have been rounded to the nearest hundredth with ≤0.01 for any values less than 0.01. Column headings at the top define the content of each column. Response categories for each characteristic are indented below the corresponding characteristic.

**Table 5 nutrients-13-01182-t005:** Associations between milk consumption tendencies and lung function measurements in NHANES participants reporting past or current asthma.

Milk Consumption Tendencies	FEV_1_		FVC		FEV_1_%	FVC%			FEV_1_/FVC	
(*n* = 1538)	β ^1^ [SE] ^2^	*p* ^3^	β ^1^ [SE] ^2^	*p* ^3^	β ^1^ [SE] ^2^	*p* ^3^	β ^1^ [SE] ^2^	*p* ^3^	β ^1^ × 10^3^ [SE × 10^3^] ^2^	*p* ^3^
Regular Milk Drinker (REGMLK vs. Never)		0.38		0.06		0.38		**0.05**		0.46
Lifetime	21.49 [51.0]	*0.68*	67.81 [62.0]	*0.28*	0.39 [1.59]	*0.81*	1.34 [1.34]	*0.32*	−9.72 [7.99]	*0.23*
Variable/Sometimes	60.61 [51.7]	*0.25*	127.282 [56.8]	*0.03*	1.64 [1.38]	*0.24*	2.62 [1.08]	***0.02***	−7.83 [7.64]	*0.31*
30-Day Milk Consumption Frequency (MLKCONS30 vs. Never)		0.26		0.45		0.25		0.56		0.47
Rarely	106.54 [64.4]	*0.10*	116.10 [75.3]	*0.13*	2.93 [1.62]	0.08	2.23 [1.55]	0.16	2.25 [9.31]	*0.20*
Sometimes	44.76 [64.0]	*0.49*	69.63 [67.8]	*0.31*	0.94 [1.60]	*0.56*	1.31 [1.25]	*0.30*	−1.75 [7.37]	*0.81*
Often	21.08 [60.9]	*0.73*	60.32 [69.5]	*0.39*	0.48 [1.75]	*0.78*	1.17 [1.45]	*0.42*	−8.77 [6.72]	*0.81*
Regular 30-day Exclusive Type Milk Consumption (TYPE30 vs. Never)		0.08		***≤0.01***		0.22		***≤0.01***		0.54
Whole	−78.40 [51.6]	*0.14*	−76.80 [62.9]	*0.23*	−1.71 [1.47]	0.25	−1.22 [1.27]	*0.34*	−9.56 [8.95]	*0.29*
2%	67.54 [56.7]	*0.24*	108.14 [64.8]	*0.10*	1.02 [1.45]	*0.48*	1.44 [1.29]	*0.27*	−3.58 [5.54]	*0.52*
1%	114.60 [78.6]	*0.15*	144.53 [90.8]	*0.12*	4.24 [2.36]	*0.08*	4.20 [2.14]	*0.06*	0.46 [9.63]	*0.96*
Skim	86.53 [72.2]	*0.24*	138.04 [79.9]	*0.09*	2.07 [1.75]	*0.24*	2.76 [1.34]	*0.05*	−6.96 [8.21]	*0.40*
Other	83.76 [92.1]	*0.37*	51.15 [93.7]	*0.59*	0.26 [2.51]	*0.92*	−1.21 [2.25]	*0.59*	9.10 [13.5]	*0.50*
2+ types	15.82 [98.7]	*0.87*	21.34 [119.7]	*0.86*	2.00 [3.10]	*0.48*	1.73 [2.83]	*0.54*	1.37 [10.7]	*0.90*

^1^ Regression coefficient; Each regression model shows milk consumption adjusted for the following confounding factors: gender, race, age, education, poverty level, BMI group, smoking status, mineral dust exposure in the past, and natural dust exposure in the past. ^2^ Standard Error; Due to small numbers of regression coefficient and standard error, values were displayed after multiplying by 10^3^. ^3^
*p*-values ≤ 0.05 are bolded and have been rounded to the nearest hundredth with ≤0.01 for any values less than 0.01; data rounded to the nearest hundredth (greater than or equal to 0.005 rounded up) with ≤0.01 for any values less than 0.01; non-italicized *p*-values (top *p* value for each variable) represent Wald-type/overall *p*-values. Column headings at the top define the content of each column. Response categories for each characteristic are indented below the corresponding characteristic. Definitions: forced expiratory volume in one-second (FEV_1_); forced expiratory volume in one-second (FVC).

## Data Availability

The NHANES dataset is publicly available online, accessible at cdc.gov/nchs/nhanes/index.htm.

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
