# Peer review of "Milk Consumption and Respiratory Function in Asthma Patients: NHANES Analysis 2007–2012"

_nutrients, 2021, doi:10.3390/nu13041182_

Round 1

Reviewer 1 Report

The manuscript aims to address if milk consumption has an impact on pulmonary lung function related parameters using a large population based cross-sectional study design and focusing in all population and in those with self-reported asthma. Although interesting results were found, these should be interpreted with caution and some concerns should be addressed, namely the need to better define asthma related outcomes, frequently it is used the concept of active asthma as the same as current asthma (which are not the same- is there any data regarding use of medication, how was this defined?). Additionally, there should be also clearly recognized the impact of cross-sectional design in the data interpretation. There is a need to focus on those variables that clearly show a direction and not only a trend, the highlight of p-values between 0.05 and 0.1 might be misleading of the importance of the data found.

The following topics should be addressed

  • The aim in the abstract section does not correspond to the study design aim (that is stated at the end of introduction section line 57 to 60), this study focus in lung function parameters both in all population and in those with self-reported asthma, not in “altering individual risk for asthma” – this aim would not be possible with a cross-sectional design.
  • The abstract should be organized accordingly to data presentation, first focusing in the all population, then specified in those that self-report asthma and comparing those with current asthma symptoms.
  • The abstract conclusion is also misleading it is not possible with a cross-sectional design to establish that dairy milk consumption has a clear protective effect on pulmonary health and asthma, large prospective cohort and RCT studies are needed to establish that, nevertheless the results are interesting
  • Why was not used the FEV1/FVC % ratio, this is a particularly important outcome used on the diagnosis of obstructive and restricted diseases.
  • Why was used the data on volumes and on predicted percentage for FEV1 and FVC, the last might be more useful as it is based on specific population details (as stated in 86 to 90)
  • There is a clear definition on the variable used for milk consumption frequency, although it should be specified also in the methods the meaning of whole milk (regular mil), and that 2% milk is reduced fat, 1% low fat and that skim milk is a fat free in the methods section (lines- 69 to 75).
  • No clear definition is present for the primary outcome, how was self-reported asthma analyzed, which questions were used to define? There is a need to include a table regarding the definitions used and also promote consistency on the terms applied. For example in table 2 “Yes still have asthma” is used, however in the results it is presented “history or current asthma”, in table 4 are current asthma is stated, however during the manuscript the concept of “active asthma” (line 187) is also introduced. Active asthma is a different concept of current asthma.
  • Do you have any information regarding the medication used, asthma control symptoms, this will be interesting to relate with the <30 day milk consumption.
  • There were specific confounders used on the models, this concepts should also be more clearly defined, namely cut-off for poverty index used ( table 2 it is stated poor, nearly poor and not poor- which cutt offs were used?), how was defined previous work exposure to mineral dust and natural dusts—perhaps a supplementary table regarding these definitions might be useful
  • In which previous literature was based to support the choice of these potential confounders and which variables had a significant impact in the multivariate analysis, if no impact was seen why include them in the analysis?
  • If P<0.05 was considered statistically (line 109), I consider it is confusing to highlight in the tables in bolded and italicized those that are between 0.05 and 0.1.
  • Table 2 should be re-organized, no SD is described for age. The significant values report to which comparison? Total population or between groups (No history vs history of asthma; history vs current asthma; with the all population?)
  • A flow chart regarding the patient selection and inclusion in each groups might be useful
  • Table 2. Why is 95%Cl presented in some variables, namely in natural dust and not in the other variables?
  • Table 2 Needs a better organization the data presentation perhaps in the “Lifetime Regular Milk Drinker” use “Lifetime Milk “- subdivide with Regular drinker, Never been, Sometimes
  • What is include/defined as other milk only- it should be specified, as it is relevant for results interpretation
  • Results Section should be organized to improve comprehensibility, first specify lung function in all participants, then approach those with “current asthma” in all participants, please define if it was used the 1542 individuals or only those with “Still asthma”/Current/Ative (n=825)?
  • Figure 1 what is defined as reporting active asthma, was all population compared only with those 825 individuals that “Still have asthma”?
  • In the discussion section it should be addressed why in patients with asthma history (current or past) there was only an association found with variably being a milk drinker, does the type of milk processing might have an impact? (raw versus boiled milk?)
  • It should also be discussed if the values found in lung function parameters regarding this variation might be clinically significant
  • Instead of discussing the “milk-mucus theory” a clear myth it should be further discussed if milk processing and the origin of milk, like farm milk as stated in line 307 might have a potential impact in the results
  • The limitation section should address the cross sectional design as a potential limitation and how we could overcome this limitation with future studies designs
  • With this cross-sectional design it is not possible to produce a definitive statement that those who consume dairy products have an improved pulmonary function, please reformulate
  • Further it is not possible to state that dairy products consumption reduce incidence of asthma symptoms, as only current asthma was studied and not specific symptoms, further studies will be needed to address this question

Reviewer 2 Report

While my interpretation of the overarching research question here (is dairy intake associated with better respiratory function overall and in particular for those with asthma?) is indeed interesting, albeit perhaps too broad, there are a number of issues in the manuscript that need attention. These include what I interpret as an unclear, and perhaps even changing, research question throughout the manuscript, which needs to be more clearly established and maintained throughout the work. There are also needs for more context and implications for your study, more work on the underlying causal model (without which, the reader cannot confidently interpret your results), and more focus on practical/clinical significance (e.g., it’s not clear to me how “markedly improved pulmonary function” on Line 359 has been evaluated or established) and less on statistical significance. The reporting of effect sizes and CIs alongside p-values is inconsistent throughout the Results (sometimes just p-values, other times p-values and effect sizes, rarely all three), and these need to be interpreted in the Discussion. Overall, I think that there is considerable work needed to tighten up the manuscript if it is to tell a compelling story, but at the same time, I think that with effort this would be possible. I strongly suggest working through the STROBE and STROBE-nut checklists and their explanations as these will address several of my concerns.

The abstract needs to be improved by adding effect sizes (ORs) and their associated 95% CIs. P-values are of no use for the reader wondering if these associations have any clinical/practical significance. In order to interpret the results in the abstract, the reader also needs to know what the results on Lines 18–26 mean, i.e. are these unadjusted, if not, what were they adjusted for? Confounding is a substantial, arguably the most substantial issue, with observational research and nothing can be interpreted with confidence without knowing something about the causal model underpinning your statistical analyses. I appreciate that it can be difficult to establish this within the restrictions of an abstract, but as much information as possible should be presented to the reader there. Similar comments apply to parts of the Results section also, although more information is presented there.

A question some readers will have when they are first looking at the abstract is why you are looking at both actual values and percent predicted values for FEV1 and FVC; another is why are you not looking at their ratio (the only mention I could see of the ratio was on Line 85)? For the first part of this, while you could look at one (actual or PP) as the primary outcomes and the others as secondary/sensitivity analyses, looking at both provides little information and only creates problems when the results are not consistent (this is also an issue/consequence of the focus in the abstract of reporting only p-values and there will not be enough space there to report the point estimates and CIs for all outcomes). You need to establish which is your main question and why. This cannot be based on the pattern of results and needs to be determined as if prior to these, with an appropriate explanation given to the reader. For the second part, the ratio is crucial to distinguish obstructive and restrictive patterns. If there is a reason for not looking at the ratio here, this doesn’t need to be included in the abstract, although as a reader, I would be wondering at that stage, but it will need to be very clearly explained, perhaps around Line 85.

By the end of the Introduction (Lines 31–61 here), the reader should understand what is already known, what remains unknown, and appreciate how the authors intend their work to address the gap(s) in our knowledge. Surprisingly, the only mention of asthma in the introduction before the final (aim) paragraph is on Line 34 which supplies just two references to the claim that asthma responds to diet. I’m not sure that these two references justify the second sentence in the abstract (Lines 14–15, “Many studies have demonstrated the protective effects of dairy products in immune-related diseases”), which comes just before the aim sentence there and I suggest some careful searching of the literature on this topic. I’d also like to see references provided for all claims in the introduction (e.g. those about current treatments for asthma, and the contributions of dairy to the diet). While you should use only as many references as are needed, five references (one for NHANES itself) is a very small number for an introduction, particularly given the breadth of this topic. The introduction could perhaps start by setting out the burden of asthma (this could include the costs of exacerbations and the relationship between asthma and COPD), explain the potential role of dietary factors in asthma, and identify what is already known here, before setting out why we should be interested in whether milk consumption is associated with respiratory function amongst asthmatics. The lack of information and references to lung function growth and decline and COPD were surprising to me.

In terms of outcomes, it was not immediately clear what “asthma prevalence” on Line 77 (or “asthma” on Line 102) would mean in a sample of asthmatics (Lines 19–20: “decreased likelihood of reporting active asthma among asthmatic participants”, and I initially wondered if you meant exacerbations or something else?). Nor is it clear what “altered lung function” (Lines 77–78) would mean if spirometry is being assessed at a single point in time per participant. The focus on pre-bronchodilator rather than post- values (Lines 88–91) cannot be justified by the number of participants with data. These two outcomes would each be addressing distinct research questions. Which is the precise research question of interest in your manuscript? As mentioned above, it seems odd to me that you haven’t looked at FEV1/FVC (Lines 102–103) and it’s not clear to me why you look at both actual and PP values.

The causal model here needs further justification and additional consideration of potentially unmodelled confounding. While it’s pleasing to see that the confounders were selected a priori (Line 92), the STROBE checklist makes it clear that these variables also need to be justified. What are your sources in the literature (Lines 92–93) for each of these variables? You will also need to think about how these variables are inter-related. For example, it’s not clear to me how occupational exposures would meet the classical definition of confounding here. While this could be a competing exposure, how would this be related to milk intake (and note that education and poverty are already in the model, along with gender and race, so it cannot be a surrogate for these)? Note that “gender” (e.g., Line 93) is not the same thing as “sex”. Which one do you think is important here? Why are you including “gender” and race (Line 93) in all models when some of your outcomes are already standardised for these? What would these variables do in such models? Is there the possibility of reverse causality between occupational exposures and respiratory function? The same issue arises around the exposure:outcome associations, as you allude to on Lines 332–341, which does not require an actual physiological association to be induced. There are many other questions that I have about these variables and I suggest drawing up a directed acyclic graph to establish the various roles of measures listed on Lines 93–103, whether they be exposures, outcomes, mediators, confounders, moderators, or competing exposures, and then carefully checking that each link in the model can be justified (preferably through citations to the literature or clear arguments based on the same). You could, if you preferred, present this textually or in a tabular format, but the information is needed.

A little more information on the statistical analyses are warranted. How much data was missing (asthma status is covered on Line 124, but not other variables as far as I could see), how was this addressed, what model diagnostics were used, how was multiplicity considered? Related to the causal model discussed above, was effect modification considered? These points all need to be explained to the reader in the statistical methods. Given the number of statistical tests performed, partly because of both actual and percentage values, there is what I would regard as over-interpretation of the findings, including of non-statistically significant results.

Table 2 appears to be a descriptive table and confidence intervals are not descriptive (I suspect that these were included due to the survey analysis, you’ll need to think about how to describe the data in this context). If you want to highlight differences between the three asthma groups, these need to indicate pairwise differences and overall p-values alone are insufficient—in particular p-values cannot be reduced to asterisks. Line 131 refers to “univariate”, where it seems “univariable” is intended. The tests used for any comparisons also need to be clear in the table (including its notes) and this will also require identifying covariates included in models. Note that there seems to be a typo for the overall column and each of the rows for sometimes/varied lifetime milk, and for 2+ types of milk. The lifetime milk drinking categories are not in order.

The study sometimes makes strong causal conclusions (e.g. Lines 357–360: “…these results indicate that individuals with asthma who consume dairy products, and specifically low-fat dairy products, will have markedly improved pulmonary function and statistically significant decrease in incidence of symptoms.” seeming to suggest an almost clinical approach to asthma management). These conclusions cannot be justified from observational data unless you can convince the reader that your causal model has effectively ruled out confounding, etc., a task that I would consider effectively impossible here. What you could conclude, perhaps, is that there is evidence for an association that could justify further study in other populations and, maybe, clinical trials will follow if the evidence can be replicated. I strongly suggest going through the manuscript and examining the text for overly causal statements (e.g. Line 15, “role…in altering individual risk” seems too strong here, as does Line 27’s “protective effect” [notwithstanding the “associated with”].) Don’t use “increase[d]”, “reduc[ing]”, “improve[d]”, or related terms unless you are referring to longitudinal changes.

Moving the summary of the results to the start of the first paragraph of the Discussion and drawing more attention to patterns and exceptions in these many analyses would be helpful. I feel that the Discussion needs more on study limitations around confounders not included here and reverse causation. It could also be improved through more on study implications for policy, practitioners, and future research. I appreciate that some of this is covered, e.g. Lines 345–351, but this could be expanded and made more concrete.

Finally, some care with the writing is needed—there were several typos and instances of “non-academic” language that careful proof-reading and editing should resolve.

Some specific comments are below:

Line 13: This (“The prevalence of asthma has been on the rise…”) is perhaps a little too vague. Could you qualify this, e.g. since when, how much?

Line 15: I know you say “potential role” but “altering individual risk” is strong causal language, more than I think can be justified from observational data here. As noted above, the manuscript needs to be carefully checked for this.

Line 18: I think that readers of the abstract will want to know (at least) the sample size, the number with asthma, and the number consuming dairy milk before you move on to the regression results.

Line 18: Subscript “1” in “FEV1” (here and elsewhere, c.f. Line 63).

Line 18: The reader might assume percentages of predicted here (from “as percentages”), but you should be explicit about what these percentages are of.

Line 19: Readers will want to know what “regular consumption” means here in the abstract.

Line 20: Readers will want some idea of what “active asthma” means here.

Line 20 and elsewhere: 3 decimal places is enough for p-values, with “p<0.001” for smaller values.

Line 23: “protective” is causal language.

Line 25: “improved” is causal and implies longitudinal changes.

Lines 26–28: This seems a much too strong interpretation for observational data. I recommend looking at a more cautious presentation and also adding some of the implications (for one or more of policy, practitioners, and researchers) here. What should the reader of your abstract see as the next step from your findings?

Lines 109 and 113: Presumably “two-sided”. It’s not clear why this is stated twice.

Lines 109–110: This seems repeated. Note that CIs are not descriptive (nor are standard errors) and there is no indication that the methods have transitioned to the inferential analyses here.

Lines 119–121: These appear to be left over instructions to authors.

Lines 131, 134, etc.: Note that “univariate” means a single dependent variable not a single independent variable. On Line 138, you mean “multivariable” (also elsewhere).

Lines 134–137: These are not associations with “asthma diagnosis” as far as I can tell, as these are over three levels of asthma (no history, history without current, and current). You’ll need to explain these more clearly in the context of these three levels, including effect sizes not just statistical significance.

Line 140 and elsewhere: 0.05<p<0.10 is not a “trend” (see the many articles on this and related terms in the literature; the first example I found with Google was: https://dx.doi.org/10.1001/jamaoncol.2018.4524). Similar objections apply to “borderline” (Line 167) and other such terms. If you are using hypothesis testing, don’t over-interpret values that fall outside of the region of significance.

Round 2

Reviewer 1 Report

The authors addressed all coments and suggestions and improved the manuscript accordingly. In some Tables, namely table 2, there are minor text editing, some "," are missing in the CI 95%, perhaps use the same decimal cases for all variables will be better.  

Reviewer 2 Report

Thank you for your responses to my queries and for your revisions which I think have addressed most of these. I have a smallish number of mostly minor comments below for your consideration.

I’m sorry for nit-picking, but the “e.g.” in the graphical abstract for “PULMONARY FUNCTION” (left-most panel) seems to be an exhaustive list and so could be deleted, or replaced with “i.e.”

In the abstract, I’d include “potential” in “a priori POTENTIAL confounders” as it’s not immediately obvious that some of these could affect dairy consumption (from the graphical abstract, readers might wonder about smoking status and pulmonary exposures in particular, but these could be weak–moderate proxies for other variables that would be potential confounders, such as health consciousness and SES). You use this wording in Section 2.3 later. It’s fine to include non-confounders in regression models, and well worth including competing exposures to improve the precision of estimates, as along as the variables are not on the causal pathway (and assuming that total effects are of interest rather than direct effects).

In the abstract, I’d like to see CIs in “(β:1.81; p= 0.020 and β:1.27; p= 0.046)” if the word count allows (as you do a couple of lines below). For both this result and the following one for varied-regular milk consumption, the reader will need to know the reference category used.

I think that some respiratory physicians will question the statement that asthma increases the risk of COPD, particularly given asthma-COPD overlap syndrome, but I can understand you not wanting to get into that here.

I’d be careful about using “effective” (“are effective in managing symptoms”) and “efficacy (“this efficacy is dependent on”) as if they were synonyms as these are two different things: the pragmatic effect allowing for real-world conditions (including non-compliance) versus the response under ideal conditions (without non-compliance). I’d probably just delete “efficacy” here.

For the last sentence in the opening paragraph, I’d quality “individuals with MODERATE TO SEVERE asthma are reliant” (or something else that you agree with) as more than a “semblance of normalcy” is certainly achievable for some milder cases without intervention.

In Section 2.3, you say “overweight: 25-30, and obese: >30” but the WHO (https://www.who.int/news-room/fact-sheets/detail/obesity-and-overweight) appear to say “obesity is a BMI greater than or equal to 30” (i.e., making 30 exactly obese and not merely overweight). I appreciate that BMI values will never be exactly 30, but your definition here should be consistent with the WHO’s. I wonder if you could add a reference to this sentence also (starting “The World Health Organization has standardized”).

For the statistical methods (Section 2.4), I wouldn’t include confidence intervals as descriptive statistics (these are inferential, providing evidence about the population rather than describing the sample). I appreciate the difficulties of providing SDs for survey-adjusted data, so perhaps you could delete the first reference to “95% confidence intervals”?

For the model diagnostics (and thank you for adding these details), I’ll pedantically suggest adding “diagnostic plots showed NO EVIDENCE AGAINST normality, linearity and equal variances” as you’re not showing/testing for any of these in the exact sense but rather looking for evidence of meaningful departures from these. After this, you give the level of significance used, and while this is normally (and almost always should be) “two-sided”, it is worth mentioning this, e.g. “A TWO-SIDED significance cutoff was set to p < 0.05.”

My more important question is around the use of “ordinal logistic regression”. If you mean with proportional odds, this should have been tested (and so would become part of the model diagnostics); similarly if you mean generalised ordinal logistic regression (with some coefficients not constrained to be identical over categories, here no/not current versus current and no versus not current/current asthma). If you mean nominal logistic regression, this should be made clear here and under Table 1. Some results later seem to be from binary logistic regression, which wasn’t explicitly mentioned here as distinct from the ordinal models.

For Table 2, the heading “Overall p-values” had me expecting Wald-type p-values, one per variable rather than one per (non-reference) level, which seems to be the case in Table 3 (headings there were “p-value” and this might need to be labelled more clearly to distinguish these from the non-reference level p-values) and I think the tables could be made consistent in this regard. Table 4 then takes a third approach, giving only (what appear to be) Wald-type p-values. In any case, it seemed odd to me that “Other race” would be the reference for race in Table 2. Was there a reason for this choice rather than White (non-Hispanic) (as used in Table 4 and a more natural choice for me)? The rows with the %s and CIs for “Any type milk consumption” don’t seem aligned with the labels. The same applies to “Exclusive type milk consumption”. The mean ages are misaligned (for history of asthma). Some of the p-values seem also misaligned (e.g. “Mine dust”). Table 4 also seems to change some other reference categories compared to Table 2, specifically gender and regular milk drinker.

In Section 3.2 it might be worth clarifying that “FEV1 (p= 0.004)” is for a Wald-type p-value. The reference groups for the often and 1% milk results could be added to the text to save the reader needing to check the table for these. See also “current asthma status (p= 0.006)” in Section 3.4, and elsewhere.

The results in Section 3.3 give a single OR for each model so these are either from binary logistic regression (whereas the statistical methods only mention ordinal logistic regression) or from an ordinal model assuming proportionality, where this OR would, I’m assuming, be for yes versus never/not current. Either way, could this be made clear? Table 4/Section 3.4 are clearly using binary logistic regression and not ordinal.

Sorry, I’m too caution to say “impact” without qualification, e.g. “appreciate the POTENTIAL impact of dairy consumption on pulmonary health” at the end of the second paragraph in the Discussion.

“improved” in “In this cohort, milk consumption overall was associated with improved pulmonary function” (Page 14) suggests longitudinal changes to me and I suggest “higher” instead.
